# Characterization of the In Vitro and In Vivo Efficacy of Baloxavir Marboxil against H5 Highly Pathogenic Avian Influenza Virus Infection

**DOI:** 10.3390/v14010111

**Published:** 2022-01-08

**Authors:** Keiichi Taniguchi, Yoshinori Ando, Masanori Kobayashi, Shinsuke Toba, Haruaki Nobori, Takao Sanaki, Takeshi Noshi, Makoto Kawai, Ryu Yoshida, Akihiko Sato, Takao Shishido, Akira Naito, Keita Matsuno, Masatoshi Okamatsu, Yoshihiro Sakoda, Hiroshi Kida

**Affiliations:** 1Shionogi & Co., Ltd., Osaka 561-0825, Japan; keiichi.taniguchi@shionogi.co.jp (K.T.); yoshinori.ando@shionogi.co.jp (Y.A.); rs-km63@mail.doshisha.ac.jp (M.K.); shinsuke.toba@shionogi.co.jp (S.T.); haruaki.nobori@shionogi.co.jp (H.N.); takao.sanaki@shionogi.co.jp (T.S.); takeshi.noshi@shionogi.co.jp (T.N.); makoto.kawai@shionogi.co.jp (M.K.); ryu.yoshida@shionogi.co.jp (R.Y.); akihiko.sato@shionogi.co.jp (A.S.); akira.naito@shionogi.co.jp (A.N.); 2Department of Disease Control, Faculty of Veterinary Medicine, Hokkaido University, Hokkaido 060-0818, Japan; okamatsu@vetmed.hokudai.ac.jp (M.O.); sakoda@vetmed.hokudai.ac.jp (Y.S.); 3International Institute for Zoonosis Control, Hokkaido University, Hokkaido 001-0020, Japan; matsuk@czc.hokudai.ac.jp (K.M.); kida@vetmed.hokudai.ac.jp (H.K.); 4Global Station for Zoonosis Control, Global Institution for Collaborative Research and Education (GI-CoRE), Hokkaido University, Hokkaido 001-0020, Japan

**Keywords:** baloxavir marboxil, H5N1 highly pathogenic avian influenza virus, viral replication, inhibition, lung inflammation, combination therapy, oseltamivir

## Abstract

Human infections caused by the H5 highly pathogenic avian influenza virus (HPAIV) sporadically threaten public health. The susceptibility of HPAIVs to baloxavir acid (BXA), a new class of inhibitors for the influenza virus cap-dependent endonuclease, has been confirmed in vitro, but it has not yet been fully characterized. Here, the efficacy of BXA against HPAIVs, including recent H5N8 variants, was assessed in vitro. The antiviral efficacy of baloxavir marboxil (BXM) in H5N1 virus-infected mice was also investigated. BXA exhibited similar in vitro activities against H5N1, H5N6, and H5N8 variants tested in comparison with seasonal and other zoonotic strains. Compared with oseltamivir phosphate (OSP), BXM monotherapy in mice infected with the H5N1 HPAIV clinical isolate, the A/Hong Kong/483/1997 strain, also caused a significant reduction in viral titers in the lungs, brains, and kidneys, thereby preventing acute lung inflammation and reducing mortality. Furthermore, compared with BXM or OSP monotherapy, combination treatments with BXM and OSP using a 48-h delayed treatment model showed a more potent effect on viral replication in the organs, accompanied by improved survival. In conclusion, BXM has a potent antiviral efficacy against H5 HPAIV infections.

## 1. Introduction

Since 1997, human infections caused by H5N1 highly pathogenic avian influenza viruses (HPAIVs) have been reported in Africa, Asia, Europe, and the Middle East with high mortality and morbidity [1,2]. The spread of H5N1 HPAIVs in poultry populations also increases the risk of human infections. Therefore, the emergence of HPAIVs, which is not only evident in poultry, but also in humans, is a major public health problem [3]. H5N6 HPAIVs sporadically crossed the species barrier and have been reported to infect humans since 2014 [4]. In 2021, human infection caused by H5N8 HPAIVs was first reported in Russia [5]. Additionally, human influenza H5N1 virus infection has also been reported to cause severe pneumonia, resulting in the death of most patients due to progressive respiratory failure with acute respiratory distress syndrome (ARDS) [6,7]. H5N1 viruses infect human alveolar epithelial cells (AEC) or alveolar macrophages in the lower respiratory tract [8]. The loss of the alveolar barrier function due to AEC disruption has also been reported to result in severe pneumonia [9]. Levels of proinflammatory cytokines and chemokines, such as interleukin (IL)-6 and monocyte chemoattractant protein (MCP)-1 (also known as CCL2), are a hallmark of H5N1 virus infection in the lungs of patients, supporting the hypothesis that a harmful cytokine storm is central to its disease pathogenesis [10]. In a previous report, the reduction in viral titers in the lungs by anti-influenza drug treatments positively correlated with both the suppression of proinflammatory cytokine production and disease severity in an H5N1 virus-infected murine model, thereby establishing that a strong inhibition of viral replication can reduce lung dysfunction [11].

Baloxavir marboxil (BXM), which is converted metabolically to its active form, baloxavir acid (BXA), is a first-in-class cap-dependent endonuclease (CEN) inhibitor that has recently been approved for clinical use in over 30 countries in 2021 [12]. Notably, BXA exhibits broad antiviral activities against several types and subtypes of influenza viruses, including zoonotic strains in vitro [13,14,15]. However, BXA’s susceptibility to H5 HPAIVs isolated during various seasons, including H5N8 viruses that have recently been isolated, remains unknown. Moreover, compared with other anti-influenza drugs, BXM showed greater and more rapid reductions in viral load after treatment regimen initiation in clinical and in vivo nonclinical studies [16,17,18,19,20,21]. Nevertheless, due to the sporadic presentation of clinical cases, clinical assessments in H5 HPAIV-infected patients are difficult. Therefore, it is beneficial to assess the antiviral effect of BXM and its regimen using in vivo models.

For treating severe influenza cases, combination therapy using antiviral drugs with different mechanisms of action provides theoretical benefits [22,23]. Further, studies have shown that combination therapy with oseltamivir phosphate (OSP) and BXM or favipiravir (FPV) is effective in mice [17,24,25,26]. In our previous study, the antiviral activity of peramivir in clinical settings, in addition to other neuraminidase inhibitors (NAIs) against an H5N1 HPAIV clinical isolate, A/Hong Kong/483/1997 strain, was evaluated in vivo [11]. However, there is insufficient therapeutic evidence for combination therapy with BXM and OSP from infected animal models of zoonotic strains. Therefore, this in vivo study reports the antiviral activities of BXM monotherapy or BXM/OSP combination therapy against A/Hong Kong/483/1997 (H5N1).

HPAIVs are susceptible to NAIs [27], and treatment with oseltamivir has been reported to be effective in patients with H5N1 HPAIV infections [28]. However, the emergence of drug-resistant mutants (e.g., H274Y in the NA) has been detected after NAI treatment in patients with this virus strain [29,30]. These mutations were also observed during animal surveillance studies of circulating H5N1 influenza viruses and have been identified in avian and human isolates as well, including an isolate from a patient treated with oseltamivir [31,32]. Furthermore, in BXM clinical trials, Polymerase acid (PA) amino acid substitutions (e.g., I38T) in seasonal influenza virus strains were detected in some patients after drug administration [14,33,34,35]. However, no reports of emerging zoonotic strains with reduced BXA susceptibility exist. Hence, it is important to monitor and verify the BXA susceptibility of zoonotic strains in preparation for future emergences. To this end, viruses harboring the PA amino acid substitution in BXM-treated-mice were also monitored in this study.

## 2. Materials and Methods

### 2.1. Compounds

BXM and BXA were synthesized at Shionogi & Co., Ltd. (Osaka, Japan), whereas oseltamivir acid (OSA) was purchased from Toronto Research Chemicals Inc. (Toronto, Ontario, Canada). Subsequently, OSP was obtained from Sequoia Research Products Ltd. (Pangbourne, UK), peramivir trihydrate (PRV) was purchased from AstaTech, Inc. (Philadelphia, PA, USA), and FPV was supplied by PharmaBlock Sciences, Inc. (Nanjing, China).

### 2.2. Cells and Viruses

Madin–Darby canine kidney (MDCK) cells were maintained at 37 °C under 5% CO_2_ in Minimum Essential Media (MEM; Nissui Pharmaceutical) supplemented with 10% heat-inactivated fetal bovine serum (FBS), 2 mmol/L L-glutamine, 50 units/mL penicillin, 50 µg/mL streptomycin, and 0.05% sodium hydrogen carbonate. Human lung adenocarcinoma epithelial (A549) cells were maintained at 37 °C under 5% CO_2_ in Dulbecco’s Modified Eagle Medium supplemented with 10% heat-inactivated FBS, 50 units/mL penicillin, and 50 µg/mL streptomycin. The non-mouse-adapted avian influenza A/Hong Kong/483/1997 (H5N1) virus, which is a clinical isolate that is pathogenic in mice [36,37], in addition to A/ruddy turnstone/Delaware/103/2007 (H5N1), A/muscovy duck/Vietnam/OIE-559/2011 (H5N1), A/whooper swan/Mongolia/2/2006 (H5N1), A/black swan/Akita/1/2016 (H5N6), A/northern pintail/Hokkaido/M13/2020 (H5N8), A/whooper swan/Fukushima/0701B002/2021 (H5N8), and A/whooper swan/Miyagi/0402B001/2021 (H5N8) strains [38,39,40,41], were propagated in embryonated chicken eggs and harvested from virus-containing allantois fluids. Recombinant A/Hong Kong/483/1997 (H5N1) viruses (wild-type, harboring NA/H274Y and NA/N294S) were generated via plasmid-based reverse genetics [42]. Subsequently, recombinant viruses were also propagated in embryonated chicken eggs and harvested from virus-containing allantois fluids, after which infectious titers of all viruses were determined using the standard 50% tissue culture infectious dose (TCID_50_) assay in MDCK cells.

### 2.3. Viral Yield Reduction Assay

Two days before infection, MDCK cells were seeded in 96-well plates. These cells were then infected with the indicated viruses at 100 TCID_50_/well. Subsequently, infected cells were incubated at 35 °C under 5% CO_2_ for 1 h. Later, the viral inoculum was washed out, followed by the addition of the fresh medium and different concentrations of test compounds. Afterwards, the cells were incubated at 35 °C under 5% CO_2_ for 24 h, and viral titers (TCID_50_/mL) in the culture supernatants were determined in MDCK cells. The 90% effective concentration (EC_90_) was finally calculated as the concentration decreasing the viral titers in the culture supernatant to one-tenth of the untreated control values using the linear interpolation method. The mean and standard deviation (SD) values were calculated from three independent experiments.

### 2.4. Animal Experiments

#### 2.4.1. Experiment 1

Six-week-old female BALB/c mice (Japan SLC, Inc., Shizuoka, Japan) were maintained in a temperature and humidity-controlled environment. The mice were then infected intranasally with 75 TCID_50_ (31.3 times of 50% mouse lethal dose [MLD_50_]) of the A/Hong Kong/483/1997 (H5N1) virus, and treatment started immediately after viral inoculation. Subsequently, the mice were treated with BXM (0.5, 5, or 50 mg/kg/dose) twice daily (12-h interval between each dosing) for 1 or 5 day(s) through oral gavage. The dosing regimen of BXM 5 mg/kg/dose, twice daily for 5 days, is an extrapolated clinical setting from our previous mice models [19,20]. However, for the controls, a vehicle (0.5 *w*/*v*% methylcellulose) or OSP (5 mg/kg/dose [clinically-equivalent dose; 75 mg/kg/day in human [43]] or 50 mg/kg/dose) was administered twice daily for 5 days through oral gavage. The dosing volume was 10 mL/kg and was calculated based on the body weight before each dosing. The survival rates and body weight changes were then monitored throughout a 14-day period after the infection (*n* = 10/group). The viral titers in the lungs of mice at 1, 3, and 5 days post infection (dpi) were finally determined in MDCK cells (*n* = 5/group), whereas viral titers in the brains and kidneys of mice at 6 dpi were determined in MDCK cells (*n* = 5/group).

#### 2.4.2. Experiment 2

The mice were infected intranasally with 75 TCID_50_ (31.3 MLD_50_) of the A/Hong Kong/483/1997 (H5N1) virus, and treatment started 48-h after viral inoculation. The mice were treated with BXM (5 or 50 mg/kg/dose) twice daily for 5 days through oral gavage. For the controls, the oral gavage method was used to administer the vehicle or OSP (10 mg/kg/dose) twice daily for 5 days. Combination therapy was then performed with BXM (5 mg/kg) and OSP (10 mg/kg) twice daily for 5 days through oral gavage. The dosing volume was 10 mL/kg and was calculated based on the body weight before each dosing. Subsequently, the survival rates and body weight changes were monitored throughout a 21-day period after the infection (*n* = 5/group). Then, the viral titers in the lungs, brains, and kidneys of mice at 3, 5, 6, and 7 dpi were determined in MDCK cells (*n* = 5/group).

The details of the animal groups and their total sizes are described in Appendix A. During all experiments, the animals were housed in self-contained units (Tokiwa Kagaku, Tokyo, Japan) at the BSL-3 and ABSL-3 facilities of the Faculty of Veterinary Medicine, Hokkaido University, Japan. Additionally, the animal experiments were conducted following the guidelines of the Institutional Animal Care and Use Committee of Hokkaido University (approval numbers 15-0063, 15-0067, 15-0068, and 18-0107). The mice were euthanized when they lost more than 30% of their body weight compared with their pre-infection weight.

### 2.5. Sequence Analysis of the PA Region in Experiment 1

Viral RNAs derived from lung homogenates of BXM-treated mice were extracted using the QIAamp^®^ Viral RNA Mini Kit (QIAGEN) according to the manufacturer’s protocol. Then, a quantitative real-time reverse-transcriptase polymerase chain reaction (RT-qPCR) was used to quantify viral RNA in the obtained samples. Subsequently, Sanger sequencing was used to conduct sequence analysis of the PA region (the PA gene of A/Hong Kong/483/1997 [H5N1] strain). All RNA samples quantified over the lower limit of quantification (800 copies/reaction) were subject to this analysis (the detail of this study is in shown in Appendix A). LSI Medience Corporation (Tokyo, Japan) conducted the RT-qPCR and Sanger sequencing.

### 2.6. Quantitative Analysis of Proinflammatory Cytokines and Chemokines

Lung samples were collected from viral titer experimental samples infected with 75 TCID_50_ of the A/Hong Kong/483/1997 (H5N1) strain at 5 dpi (experiment 1). Next, the collected lungs were homogenized, after which the levels of IL-6 and MCP-1 in the lungs were quantitatively determined using Quantikine ELISA (R&D Systems, Minneapolis, MN, USA) according to the manufacturer’s protocol.

### 2.7. Histopathological Experiments

Lungs were collected from mice infected with 75 TCID_50_ of the A/Hong Kong/483/1997 (H5N1) strain at 5 dpi (*n* = 3/group). Then, the samples were fixed through perfusion in 10% phosphate-buffered formalin. Formalin-fixed left lungs were subsequently dissected, embedded in paraffin, and sectioned. Finally, hematoxylin and eosin (HE)-stained sections were prepared and subsequently used for histopathological analysis by Sapporo General Pathology Laboratory Co., Ltd. (Sapporo, Japan).

### 2.8. Assessment of Lung Index

The body weights of mice infected with 75 TCID_50_ of the A/Hong Kong/483/1997 (H5N1) strain were measured, following which their lungs were collected at 5 dpi (*n* = 5/group). Whole lungs were then weighed, and the wet weight-to-body weight ratios of the lungs in each treatment group were calculated as previously described [44].

### 2.9. Combined Effects of BXA and NAIs In Vitro

A549 cells, the A/Hong Kong/483/1997 (H5N1) virus (319 TCID_50_/well), and compounds in serial dilutions (for BXA, 0.1-8.0 nmol/L; for OSA, 1.6-400 nmol/L; for peramivir trihydrate, 1.6-100 nmol/L) were simultaneously seeded in 96-well plates. The infected cells were then incubated at 37 °C under 5% CO_2_ for 4 days. All supplements were prepared using 2% FBS in MEM. After incubation, cell viability was assessed with an MTT reagent, as previously reported [45]. Subsequently, a data analysis for yielding isobologram plots and calculating the combination index (CI) values was conducted as previously reported [17]. In detail, CI values, under the condition that both substances were added at the concentration ratio of each EC_50_ value, were calculated using the following formula: CI = (D_A/A + B_)/D_A_ + (D_B/A + B_)/D_B_ + (D_A/A + B_ × D_B/A + B_)/(D_A_ × D_B_). The combination effect was determined according to the following criteria: CI ≤ 0.8, synergy; 0.8 < CI < 1.2, additive; 1.2 ≤ CI, antagonism.

### 2.10. Statistical Analysis

For comparing the survival periods after infection between each BXM-treated group or BXM/OSP combination group and vehicle-treated or OSP-treated groups, the log-rank test was performed. However, Dunnett’s multiple-comparison test was performed for comparisons between viral titers in mouse tissues, cytokine and chemokine levels in lung tissue and lung wet weight-to-body weight ratios between each BXM-treated group, or between the BXM/OSP combination group and vehicle-treated or OSP-treated groups at each time. All statistical analyses were conducted using the statistical analysis software SAS version 9.2 for Windows (SAS Institute, Cary, NC, USA). *p* values < 0.05 were considered statistically significant.

### 2.11. Ethics Statement

The animal experiments were authorized by the Institutional Animal Care and Use Committee of the Faculty of Veterinary Medicine, Hokkaido University (approval numbers 15-0063, approved on 1 June 2015; 15-0067, approved on 11 June 2015; 15-0068, approved on 11 June 2015; and 18-0107, approved on 23 July 2018), and performed according to the guidelines of this committee. The facilities where the animal experiments were conducted are certified by the Association for Assessment and Accreditation of Laboratory Animal Care International.

## 3. Results

### 3.1. Inhibitory Effects of BXA on H5 HPAIV Replication In Vitro

To examine the antiviral activity of BXA against H5 HPAIVs, including recently isolated H5N8 variants, in vitro drug susceptibility to BXA was evaluated. BXA showed comparable inhibition efficacy in tested viruses (the mean EC_90_ ranged from 0.7 to 1.5 nmol/L) (Table 1), compared to that of A/Hong Kong/483/1997 (H5N1) and other H5 viruses (the mean EC_90_ of 0.7 to 1.6 nmol/L), as previously reported [13]. Moreover, BXA also showed inhibitory activities against viruses harboring the NA-H274Y and NA-N294S substitution, which has been related to OSA resistance [29,46]. In contrast, OSA required approximately 14-fold or higher concentrations to accomplish similar levels of viral reduction compared to BXA. However, FPV showed comparable antiviral activity against the tested viruses, but the concentration range of FPV used remained higher than that of BXA. These results therefore demonstrate that BXA had more potent antiviral activities in vitro than OSA and FPV against H5 HPAIVs. As all tested viruses had comparable susceptibility to BXA, the A/Hong Kong/483/1997 (H5N1) strain was selected for subsequent analyses.

### 3.2. The Protective Efficacy of BXM on Lethal H5N1 HPAIV Infections In Vivo

We previously reported the antiviral activity of NAIs by using A/Hong Kong/483/97 (H5N1)-infected mice [11]. To evaluate the effects of oral BXM (prodrug form of BXA) against H5N1 HPAIVs in a lethal infection model, mice were inoculated with 31.3 MLD_50_ of the A/Hong Kong/483/1997 (H5N1) strain. Mice treated with vehicle- or clinically-equivalent doses of OSP 5 mg/kg/dose administered twice daily for 5 days died within 7 and 9 dpi, respectively (Figure 1). In contrast, supratherapeutic doses of OSP 50 mg/kg/dose administered twice daily for 5 days resulted in a 70% survival. In this setting, the survival rates of mice administered BXM 0.5, 5, and 50 mg/kg twice daily for 1 day were 20%, 100%, and 100%, respectively. All mice survived after treatment using BXM, which was administered at 5 and 50 mg/kg for 5 days. This result was consistent with that obtained from other subtype-infected mice, as previously reported [19,20]. Compared with the survival time for 14 dpi, all mice treated with BXM showed significantly prolonged survival times compared with those administered the vehicle (*p* < 0.001 in all groups) or OSP at 5 mg/kg (*p* < 0.01 [only 0.5 mg/kg twice daily for 1 day], *p* < 0.001, respectively). However, although a gradual body weight loss after virus infection was observed in the vehicle-treated mice until 6 dpi (Appendix A), BXM strongly prevented body weight loss compared with vehicle and OSP.

### 3.3. Effects of BXM on Viral Titers in Mice Infected with H5N1 HPAIVs

To examine the inhibitory effects of BXM on H5N1 HPAIV’s replication, viral titers in the lung homogenates derived from A/Hong Kong/483/1997 (H5N1)-infected mice were measured at 1, 3, and 5 dpi. Compared with the vehicle treatment, the OSP treatment decreased the viral titers in the lungs of mice at 1 dpi. However, this effect diminished in effectiveness at 3 and 5 dpi (Figure 2a). In contrast, compared with the vehicle and OSP treatments, BXM treatments significantly decreased viral titers in the lungs of mice at 1 dpi (*p* < 0.001 in all BXM-treated mice compared with vehicle-treated mice and *p* < 0.01 in all BXM-treated mice compared with OSP at 5 mg/kg-treated mice), and repeated BXM treatments at 3 and 5 dpi sustained this effectiveness. Notably, BXM treatment at 50 mg/kg strongly reduced the viral titers below the lower limit of quantification (1.5 log_10_ TCID_50_/mL), until 5 dpi. Nevertheless, although gradual increases in the viral titers for 1-day dosing of BXM at 0.5 mg/kg were observed after treatment withdrawal, the viral titers were suppressed to more than 3 or 4-logs through 1-day dosing of BXM at 5 and 50 mg/kg compared with that of the vehicle. 

Additionally, genotypic alterations in the whole PA gene of A/Hong Kong/483/97 (H5N1) derived from the lung homogenates of infected mice treated with 1 and 5-day dosing of BXM were analyzed. In these lung samples, four samples had nucleic acid changes, but three out of four samples showed a silent mutation. Another sample had isoleucine-to-valine amino acid substitution at position 127 (I127V) in the PA (Appendix A). In the H5N1 HPAIV-infected mice model, infectious viruses were detected in organs other than respiratory tissues (e.g., brain and kidney), which is proposed to be due to the spillover of high viral titers of the virus replicated in mice [47,48]. Therefore, to confirm the results, we also analyzed the inhibitory effects on viral titers in the brain and kidney homogenates at 6 dpi [47,48,49]. We observed that infectious viruses were detected in the brain of all vehicle-treated mice (Figure 2b). Infectious viruses were also detected in the brain samples of three of five 5 mg/kg OSP-treated mice, one of five 50 mg/kg OSP-treated mice, and mice subjected to the 1-day dosing of BXM at 0.5 mg/kg-treated. In contrast, no viruses were detected in the brain of any other BXM-treated groups. In the kidney, infectious viruses were detected only in the vehicle-treated mice (Figure 2c). 

### 3.4. Prevention of Inflammation in Mice Lungs following BXM Treatment

The impact of BXM treatments on inflammation in mice inoculated with A/Hong Kong/483/1997 (H5N1) at 5 dpi was subsequently evaluated. Compared to the treatment with the vehicle or OSP at 5 mg/kg, the treatment of the H5N1 virus-infected mice with BXM resulted in a significantly less pronounced production of IL-6 and MCP-1 (*p* < 0.01 in all BXM-treated groups compared with the vehicle-treated group and *p* < 0.05 in BXM at the 0.5 mg/kg-treated groups, except for 1-day dosing group, compared with OSP at the 5 mg/kg-treated groups), which resulted from the reduction of viral titers in mice lungs at 5 dpi (Figure 3a). The OSP treatment also showed a significant inhibitory effect on the IL-6 and MCP-1 production in mice lungs compared with the vehicle treatment. Subsequent analyses of proinflammatory cytokine and chemokine production in lungs, including histopathological findings from lungs treated with BXM, were performed. Acute pneumonia, including thickening and inflammatory cell infiltration of the alveolar walls, inflammatory cell infiltration within the alveoli, bronchiolar and perivascular cell infiltration, and edema, was observed in about a quarter to half of the global area of all vehicle-treated mice lungs (Figure 3b and Appendix A). These findings were mild in mice treated with OSP at 5 mg/kg for 5 days compared with those treated with the vehicle. In contrast, no abnormal findings in the lungs were observed in each lung after BXM treatment at 5 mg/kg for 5 days. In our model, the lung weight-to-body weight ratio of vehicle-treated mice was higher than that of the uninfected controls at 5 dpi (Appendix A). Likewise, the ratios of all groups treated with BXM and OSP were significantly lower than those of the groups treated with the vehicle. The results also showed that the ratio of groups treated with BXM at 5 and 50 mg/kg was significantly lower than that of those treated with OSP at 5 mg/kg (*p* < 0.05). 

### 3.5. Combination Efficacy of BXM and NAIs on H5N1 Virus Infections

BXM monotherapy results were expected to be effective with a delayed treatment onset. However, the effects of BXM were previously reported to be attenuated [20]. Therefore, the combined efficacy of BXM and OSP in H5N1 virus-infected mice was evaluated to explore the possibilities of a potent therapeutic option compared with BXM monotherapy, as previously reported [17]. First, we evaluated the combination effects of BXA and NAIs (OSA or PRV) on the H5N1 virus infection. The CPE assay was then conducted in A549 cells. The isobologram plot showed that the combination of BXA with OSA and PRV resulted in CI values of 0.19 and 0.40, respectively, indicating that BXA exhibited synergistic effects with NAIs in vitro (Figure 4) [17,50]. Next, to investigate the therapeutic effect of BXM monotherapy (5 or 50 mg/kg/dose twice daily for 5 days) or combination therapy at clinically higher doses of OSP [51] (10 mg/kg/dose twice daily for 5 days), A/Hong Kong/483/1997 (H5N1)-infected mice were treated, starting 48-h after infection. All mice treated with the vehicle and OSP at 10 mg/kg/dose-treated mice showed a gradual body weight loss after viral infection, resulting in death within 8 dpi (Figure 5). Further, mice treated with BXM at 5 mg/kg/dose showed a delayed body weight loss. However, 40% survived at 21 dpi. Additionally, mice treated with BXM at 50 mg/kg/dose showed little body weight loss and survived. In this setting, the results confirmed that BXM (5 mg/kg/dose)/OSP (10 mg/kg/dose) combined therapy suppressed the body weight loss and improved survival at 21 dpi compared with monotherapy (Figure 5). Furthermore, all BXM monotherapy groups showed significantly decreased viral titers in the lungs of mice compared with vehicle- and OSP-monotherapy groups (*p* < 0.001 in all groups) (Figure 6a). Notably, BXM and OSP combined therapies also reduced the viral titers in the lungs and tended to reduce the viral titers at 6 and 7 dpi more than those of BXM at the 5 mg/kg/dose monotherapy. Furthermore, BXM mono- or BXM/OSP-combined therapies significantly reduced the viral titers in the brain and kidney at 5, 6, and 7 dpi compared with the vehicle or OSP treatment (Figure 6b,c). In particular, no viruses were detected in the kidneys of all mice in the BXM treatment groups.

## 4. Discussion

In this study, we presented the first experimental evidence that BXA exhibits similar in vitro activity against recent H5N8 variants or H5N1 viruses harboring NAI-resistant mutants in comparison with seasonal and other zoonotic strains. We also examined the therapeutic effects of oral BXM administration in an H5N1 HPAIV-infected murine model. Previous studies have shown that H5N1 HPAIVs have an overwhelming ability to replicate and spread in primary human immune cell cultures or multiple organs of mice or ferrets, unlike seasonal strains [52,53]. Therefore, it is meaningful to clarify the inhibitory efficacy of oral BXM dosing on the reduction of the viral load in multiple organs and assess the mortality caused by H5N1 HPAIV infection in animal models. In our lethal A/Hong Kong/483/97 (H5N1)-infected mice model, a 5-day dosing of BXM at 5 mg/kg/dose was sufficient to significantly decrease the viral load in mice lungs, resulting in significant improvements in survival compared with mice that were administered the vehicle or OSP treatments.

In this model, viral replication in the brains or kidneys was detected by the spillover caused by high the viral replication in mice [47,48]. The systemic spread of the H5N1 viral strain has also been reported to be associated with disease pathogenesis in mammals, particularly for the A/Hong Kong/483/1997 (H5N1) strain, which accounts for neurotropism [36,48,54,55]. Similarly, few viruses were detected in brains and kidneys after BXM treatment, suggesting that BXA directly inhibited viral replication in tissues other than the respiratory organs. Additionally, the antiviral efficacy of BXM has first been observed in the lungs, which is proposed to contribute to the inhibition of the viral spread from the lungs to extra respiratory organs. Nevertheless, the efficacy of this dosing regimen was similar to that in mice infected with other virus subtypes, H1N1 or H7N9 [19,20]. This finding evidences that the BXM treatment, using the extrapolated clinical setting from the above mice models, has a therapeutic potential against H5 HPAIVs, with a high mortality being observed in mammals. 

BXM/OSP combined therapy is a beneficial option for reducing viral titers in the lungs and preventing death in a severe H1N1 virus-infected mice model [17]. In this study, the BXA and OSA combination showed synergistic effects in the cell cultures. Combination therapy also increased the survival of mice compared with BXM- or OSP-monotherapy in a severe H5N1 virus-infected mice model, suggesting that combination therapy was effective without antagonism for treating the H5N1 virus infection. BXM and OSP have different modes of action on the viral replication cycle. Therefore, it is considered that the combination therapy showed a strong viral inhibition effect and reduced systemic viral loads. Marathe et al. reported that combination therapy with the polymerase inhibitor FPV in addition to OSP started 96-h after the infection resulted in the complete suppression of mortality in H5N1 virus-infected mice [25]. In our study, the combination therapy with BXM and OSP started 48-h after viral infection did not completely suppress mortality. This was proposed to be due to the higher infectivity of the titers inoculated in our study compared with those used in previous studies [24,25]. Nevertheless, our results present the first evidence confirming the efficacy of combination therapy with BXM and OSP for treating severe influenza-like H5N1 HPAIV infection.

Elevated levels of proinflammatory cytokines have been detected in humans and mice infected with H5N1 HPAIV in addition to virus-induced proinflammatory cytokines, and cases of chemokine dysregulation in the lungs and other tissues, which contribute to disease severity in clinical and nonclinical cases of H5N1 virus infections [10,37,56]. The production of IL-6 and MCP-1 in the early phase of H5N1 virus infection causes an over infiltration of monocytes or neutrophils in mice lungs, thereby worsening the respiratory function [57,58]. H5N1 virus infection in humans has also resulted in severe pneumonia, thereby resulting in the death of most patients due to ARDS [59]. Similar findings were observed in H5N1 virus-infected mice [44,60,61]. Similarly, our results showed that the BXM treatment significantly decreased IL-6 and MCP-1 production and prevented increases in the lung wet weight-to-body weight ratios, which is consistent with the reduction of the viral titers in the lungs. Furthermore, the BXM treatment resulted in no histopathological changes, such as edema and hemorrhage, in mice lungs compared with the vehicle or OSP at 5 mg/kg treatments, suggesting that BXM inhibited a virus-triggered inflammation in the lungs. Additionally, in our model, although OSP showed inhibitory efficacy during the lung inflammatory responses, the survival rates of the mice treated with OSP were lower than those of the mice treated with BXM. Although the OSP treatment did not reduce the viral loads, it did reduce lung inflammation, followed by a reduction in the morbidity of seasonal influenza virus-infected mice [62]. This finding suggests that the OSP treatment immediately after viral infection suppressed lung inflammation and improved the symptoms. However, the host cytokine response inhibition is insufficient for reducing the morbidity and lethality of viral infections. A previous study has also reported that the early inhibition of viral replication is more promising than the inhibition of the cytokine response for improving host survival during a H5N1 virus infection [63]. Moreover, the polymerase complex genes of human H5N1 virus strains contribute to a high pathogenesis in mammals [64]. Some novel polymerase inhibitors (e.g., FPV and VX-787) have also shown strong efficacy for inhibiting viral replication and preventing death in H5N1 virus-infected mice compared with OSP, implying that the inhibition of polymerase activity was effective in treating H5N1 virus infection [65,66]. Therefore, BXM, which is more potent for inhibiting viral RNA transcription and thereby inhibiting virus replication, is proposed to be more suitable as a therapeutic option than OSP. 

In this study, a variant virus harboring I127V in the PA was detected in the mice after the dosing of BXM at 0.5 mg/kg twice daily for 5 days. According to a previous report, V127 in the PA was highly conserved in a recent isolate of H5N1 viruses [67]. In addition, some of the H5 HPAIVs tested in the study harbored V127 in the PA (Appendix A), and they were susceptible to BXA such as the A/Hong Kong/483/1997 (H5N1) strain. These results suggest that I127V in the PA did not impact on BXA susceptibility. In clinical settings, the emergence of drug-resistant mutants in NA has been reported after NAI treatment for H5N1 HPAIV infection [28,29]. Moreover, a few natural occurrences in H5 HPAIV strains associated with a reduced susceptibility to BXA were observed in the database [68]. Therefore, a continuous virus monitoring and susceptibility testing of BXM is required. As previously reported, BXM/OSP combination therapy reduced the frequency of the emergence of BXM-resistant strains in H1N1pdm09 virus-infected mice compared with BXM monotherapy [69]. BXM/OSP combination therapy may also be a useful option to reduce the emergence of BXM-resistant H5N1 viruses.

## 5. Conclusions

This study demonstrated that the BXA susceptibility of H5 HPAIV strains isolated over several years, including recently isolated H5N8 variants, was retained. As observed, the oral administration of BXM drastically decreased the viral loads in the lungs and extra respiratory organs. BXM administration also reduced lung inflammation and improved mortality in an H5N1 HPAIV-infected mouse model. Moreover, combination therapy with BXM and OSP can be used as a treatment option for severe influenza-like H5 HPAIV infection. We also discovered that combination therapy, compared with monotherapy, had more potent effects on viral replication in the organs, thereby improving survival when using a 48-h-delayed treatment model. Therefore, on the basis of these findings, we propose that BXM has a potent antiviral efficacy against H5 HPAIVs.

## Figures and Tables

**Figure 1 viruses-14-00111-f001:**
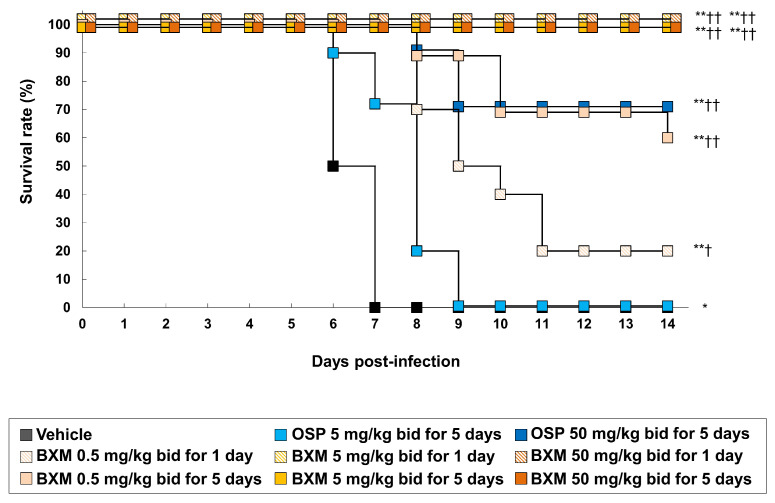
Therapeutic effects of baloxavir marboxil (BXM) on survival in a lethal infection model of mice infected with H5N1 virus. The mice were intranasally inoculated with 75 times of 50% tissue culture infectious dose (TCID_50_)/mouse (31.3 times of 50% mouse lethal dose [MLD_50_]) of the A/Hong Kong/483/1997 (H5N1) virus, and treatment was started immediately after viral inoculation (*n* = 10/group). The survival time was then monitored throughout a 14-day period after the infection. The log-rank test was performed to compare the survival time between each group (* *p* < 0.01, **, *p* < 0.001 compared with vehicle, † *p* < 0.01, †† *p* < 0.001 compared with oseltamivir phosphate (OSP) at 5 mg/kg twice daily).

**Figure 2 viruses-14-00111-f002:**
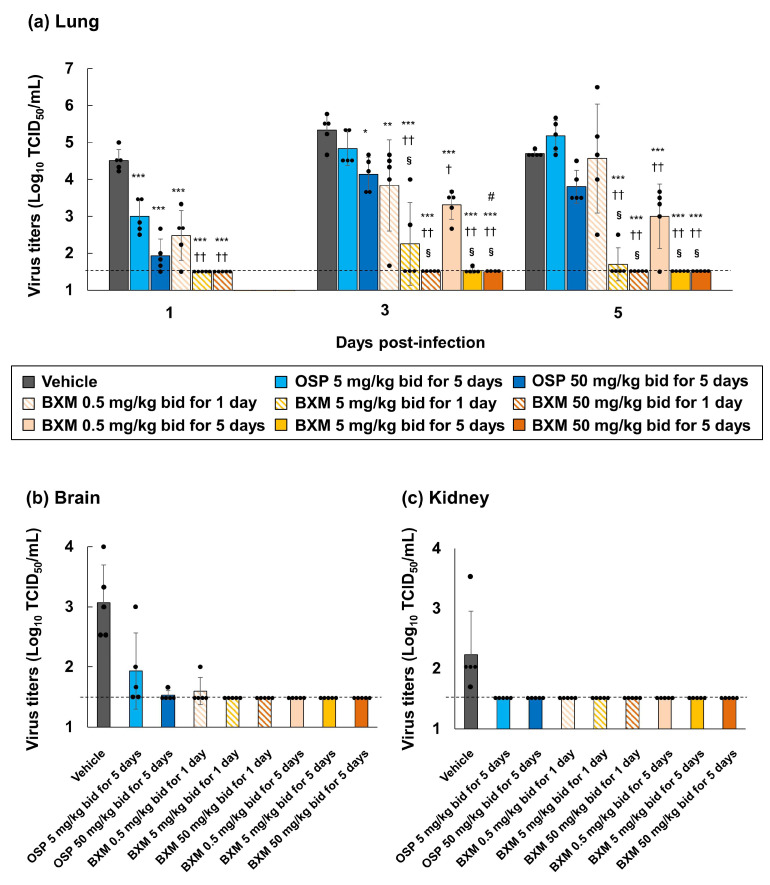
Inhibitory effects of BXM on viral titers in the lungs, brains, or kidneys of mice infected with the H5N1 virus. The mice were inoculated with 75 TCID_50_/mouse (31.3 MLD_50_) of the A/Hong Kong/483/1997 (H5N1) virus, and treatment was started immediately after viral inoculation (*n* = 5/group). (**a**) The viral titers (TCID_50_) in mice lungs at 1, 3, and 5 days post infection (dpi) measured in Madin–Darby canine kidney (MDCK) cells. (**b**,**c**) The viral titers in mice brains or kidneys at 6 dpi measured in MDCK cells. The lower limit of quantification of the viral titer was indicated using a dotted line (1.5 log_10_ TCID_50_/mL). Dunnett’s multiple-comparison test was performed for the statistical comparison of viral titers in the lungs between each group (* *p* < 0.05, ** *p* < 0.01, ***, *p* < 0.001 compared with the vehicle. † *p* < 0.01, †† *p* < 0.001 compared with OSP at 5 mg/kg twice daily. § *p* < 0.001 compared with OSP at 50 mg/kg twice daily). #, *n* = 4/group. One of 5 mice died due to an experimental error.

**Figure 3 viruses-14-00111-f003:**
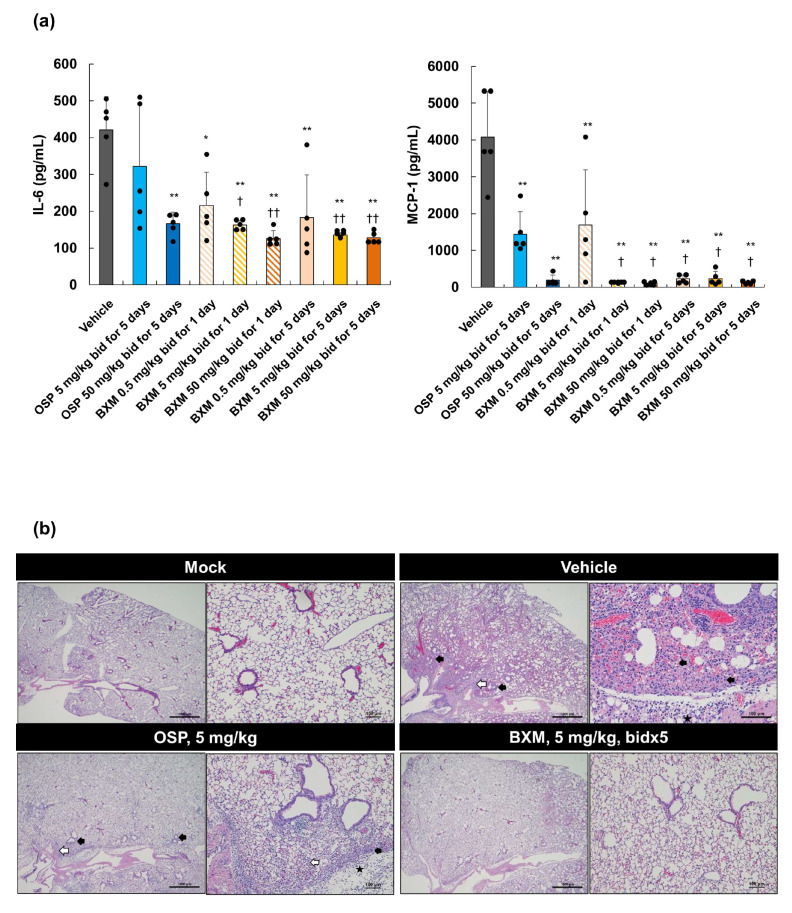
Prevention of inflammation in the lungs of mice infected with H5N1 viruses following BXM treatment. The mice were intranasally infected with 75 TCID_50_/mouse (31.3 MLD_50_) of A/Hong Kong/483/1997 (H5N1) virus, and treatment was started immediately after viral inoculation (*n* = 5/group). (**a**) interleukin (IL)-6 and monocyte chemoattractant protein (MCP)-1 production in the lungs at 5 dpi was then quantified. Dunnett’s multiple-comparison method was conducted for the statistical comparison of IL-6 and MCP-1 production between each group (* *p* < 0.01, ** *p* < 0.001 compared with the vehicle. † *p* <0.05, †† *p* <0.01 compared with OSP at 5 mg/kg twice daily). (**b**) The lungs after each dosing were collected and fixed in a perfusion, containing 10% phosphate-buffered formalin. The formalin-fixed left lungs were then dissected, embedded in paraffin, and sectioned. Hematoxylin and eosin-stained sections, which were prepared for specimen and histopathological analyses, were subsequently analyzed. The left panels of each dosing represent wide fields (2× magnification of objective), whereas the right panels of each dosing represent narrow fields (20× magnification of objective). The black arrows indicate the thickening and inflammatory cell infiltration of alveolar walls. The white arrows indicate inflammatory cell infiltration within the alveoli. The black stars indicate edema. Mock: the mice were inoculated with Dulbecco’s phosphate-buffered saline and administered vehicle (0.5 *w*/*v*% methylcellulose) twice daily for 5 days.

**Figure 4 viruses-14-00111-f004:**
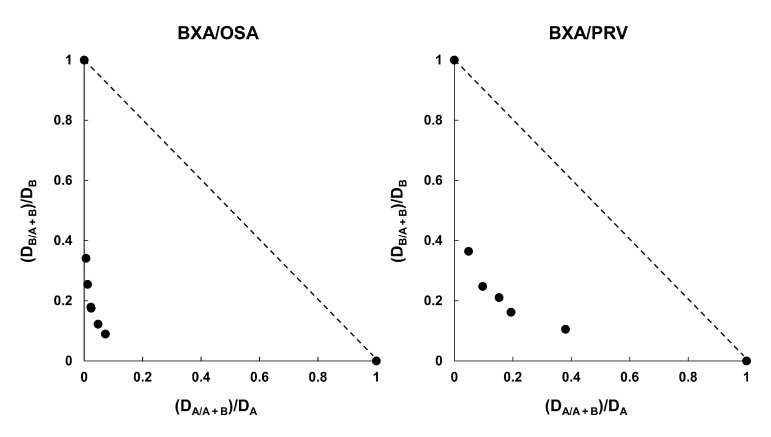
The isobologram plot of baloxavir acid (BXA) combined with oseltamivir acid (OSA) or peramivir trihydrate (PRV). The 50% effective concentration (EC_50_) of each substance alone and at a fixed concentration were determined. (D_A_/_A + B_)/D_A_ and (D_B_/_A + B_)/D_B_ were plotted on the x- and y-axes, respectively. D_A_ is the EC_50_ of substance A alone, D_B_ is the EC_50_ of substance B alone, D_A/A + B_ is the concentration of substance A; given a 50% inhibition combined with substance B, D_B/A + B_ is the concentration of substance B; given a 50% inhibition combined with substance A.

**Figure 5 viruses-14-00111-f005:**
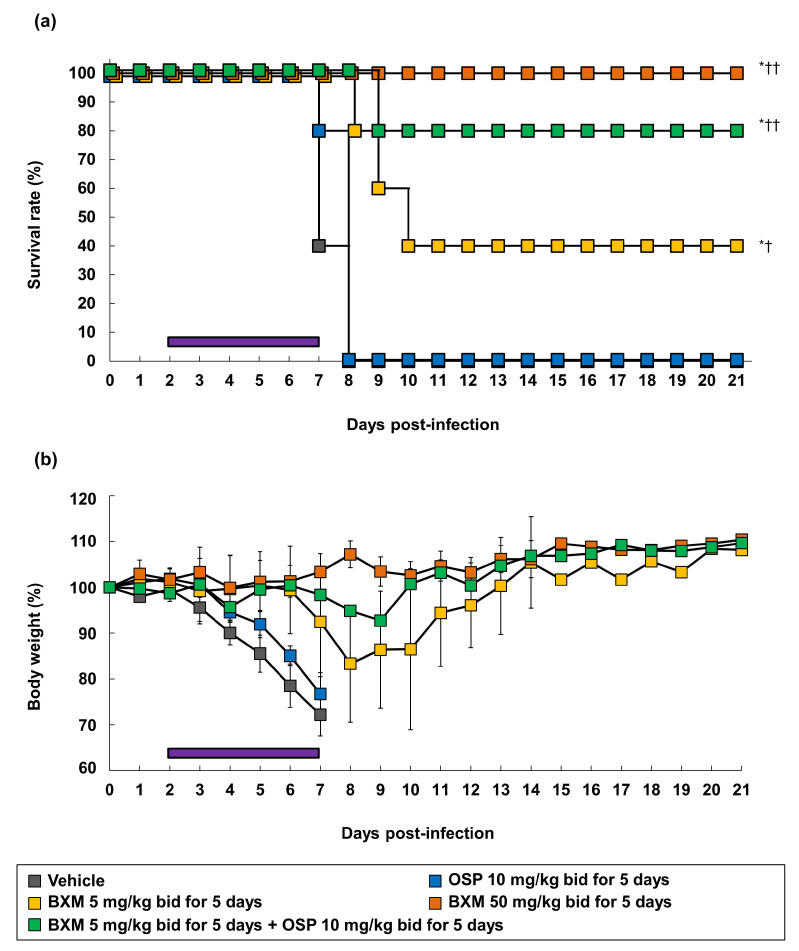
Effects of the delayed treatment of BXM, OSP, or their combination on the H5N1 HPAIV infection in the mice model. The mice were intranasally infected with a 75 TCID_50_/mouse (31.3 MLD_50_) of A/Hong Kong/483/1997 (H5N1) virus, and treatment was started 48-h after viral inoculation. (**a**) Survival time and (**b**) body weight loss were monitored throughout a 21-day period after the infection (*n* = 5/group). The purple line represents the treatment period. The log-rank test was performed for comparing the survival times between each group (* *p* < 0.01 compared with the vehicle. † *p* < 0.05, †† *p* < 0.01 compared with OSP at 10 mg/kg twice daily).

**Figure 6 viruses-14-00111-f006:**
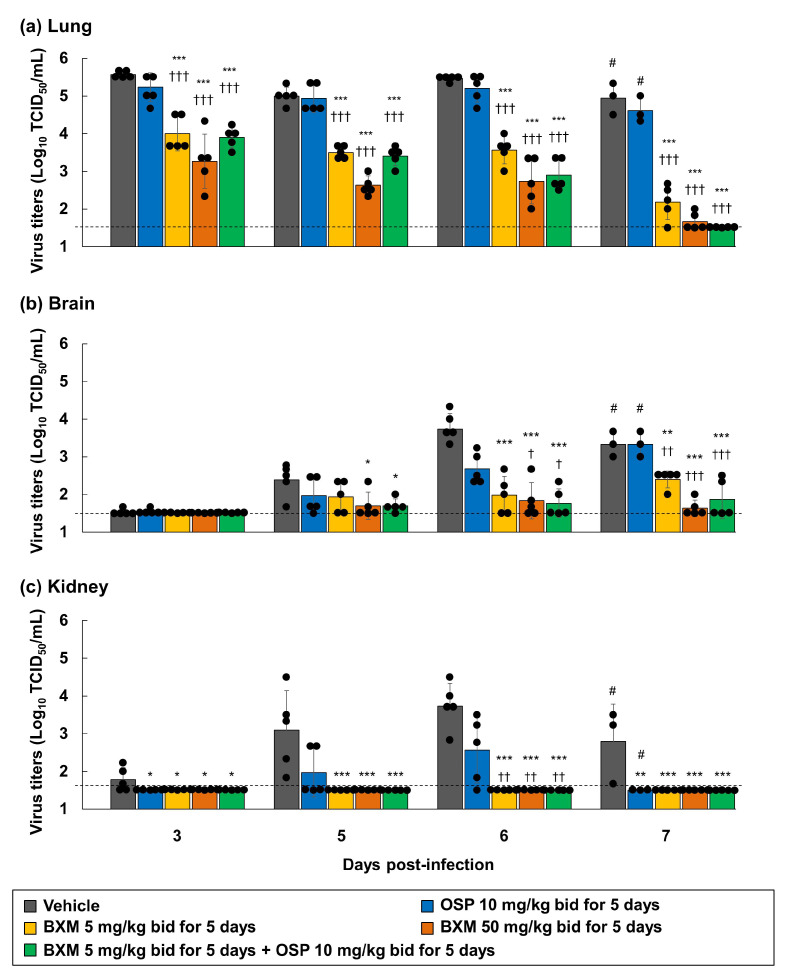
Inhibitory effects of delayed treatment of BXM, OSP, or their combination on the viral titers in the lungs, brains, and kidneys of mice infected with H5N1 HPAIV. The mice were intranasally infected with 75 TCID_50_/mouse (31.3 MLD_50_) of the A/Hong Kong/483/1997 (H5N1) virus, and treatment was started 48 h after viral inoculation. Viral titers (TCID_50_) in the (**a**) lungs, (**b**) brains, and (**c**) kidneys of mice at 3, 5, 6, and 7 dpi measured in MDCK cells (*n* = 5/group). The lower limit of quantification of the viral titer is indicated using a dotted line (1.5 log_10_ TCID_50_/mL). Dunnett’s multiple-comparison test was conducted for a statistical comparison between the viral titers in each organ group (* *p* < 0.05, ** *p* < 0.01, *** *p* < 0.001 compared with vehicle, † *p* < 0.05, †† *p* < 0.01, ††† *p* < 0.001 compared with OSP at 10 mg/kg twice daily). #, *n* = 3/group. Two of 5 mice in each group died at 7 dpi.

**Table 1 viruses-14-00111-t001:** Antiviral activities of baloxavir acid and other reference compounds against HPAIV in viral yield reduction assay.

Influenza Virus Strains	Mean EC_90_ (nmol/L) ± SD		
	Baloxavir Acid	Oseltamivir Acid	Favipiravir
A/ruddy turnstone/Delaware/103/2007 (H5N1)	1.4 ± 1.3	12.8 ± 5.7	16,927.4 ± 12,375.0
A/muscovy duck/Vietnam/OIE-559/2011 (H5N1)	1.5 ± 0.3	20.7 ± 10.9	11,689.6 ± 7333.8
A/whooper swan/Mongolia/2/2006 (H5N1)	0.9 ± 0.5	14.9 ± 1.6	13,812 ± 11,056.8
A/black swan/Akita/1/2016 (H5N6)	0.8 ± 0.5	20 ± 9.7	50,156.1 ± 69,943.5
A/northern pintail/Hokkaido/M13/2020 (H5N8)	1.3 ± 1.0	19.5 ± 11.9	16,025.5 ± 10,164.5
A/whooper swan/Fukushima/0701B002/2021 (H5N8)	1.1 ± 1.0	19.2 ± 13.9	16,052.1 ± 12,507.3
A/whooper swan/Miyagi/0402B001/2021 (H5N8)	0.7 ± 0.3	11.3 ± 2.8	6992.6 ± 1370.8
rg-A/Hong Kong/483/1997 (H5N1) *	1.6 ± 1.0	16.4 ± 11.4	26,948.7 ± 5081.5
rg-A/Hong Kong/483/1997 NA-H274Y *	3.2 ± 1.2	4054.9 ± 1295.7	31,129.5 ± 11,788.5
rg-A/Hong Kong/483/1997 NA-N294S *	1.7 ± 0.4	1291.2 ± 482.6	77,002.6 ± 2190.5

Each value represents the mean and SD for 3 independent experiments. * virus generated by reverse genetics.

## Data Availability

Not applicable.

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
