# Peer review of "Characterization of the In Vitro and In Vivo Efficacy of Baloxavir Marboxil against H5 Highly Pathogenic Avian Influenza Virus Infection"

_viruses, 2022, doi:10.3390/v14010111_

Round 1
Reviewer 1 Report
This manuscript has been greatly improved.
Reviewer 2 Report
Article is improved by including recommended information. Now, it seems suitable for publication.
This manuscript is a resubmission of an earlier submission. The following is a list of the peer review reports and author responses from that submission.
Round 1
Reviewer 1 Report
Baloxavir is a new influenza drug on the market. It was already shown that this potent cap-dependent endonuclease inhibitor is inhibited replication of human and avian influenza viruses. Since PA protein belongs to conservative proteins, it is not surprising that its activity is not restricted to one subtype of influenza virus. Authors already published similar work with H1N1 and H7N9 influenza viruses. The present manuscript shows anti-viral activity of Baloxin against H5 viruses and confirms the combination effect of BMX and OSP in mice. The article is well written and obtained results are well documented. The weak point is, that the animal experiments were not repeated, and authors could analyse data obtained from one experiment. Some partial data were already published, and the present manuscript resemble work with H7N9 avian influenza virus. I have some minor comments:
- Full name of cytokines and abbreviation in parenthesis should be used once when they are used firs time.
- MCP-1 is also known as CCL2. It will help to used both names.
- 6, 262 what is [42]- treated mice?
Reviewer 2 Report
Author has evaluated the therapeutic effects of clinically approved antiviral drug “Baloxavir Marboxil '' against highly pathogenic avian influenza strains in vivo and in vitro. Moreover, the efficacy of the drug was also tested in combination with other antivirals (Oseltamivir and peramivir) in vivo and reported the improved survival rate in infected mice after giving combined therapy suggesting the synergistic role of this drug with greater efficacy in clearing infection. Article is interesting, well written and clinically relevant. I would suggest considering it for final submission. However, I have few suggestions.
- Author has tested antiviral efficacy of BXM against various influenza strains including OSA resistant strains (NA-H274Y and NA-N294S). I am wondering why the author did not include BXM resistant strain as a control to check the inhibitory role of antivirals (Table 1) and possibly, if other antivirals (OSA & FPV) might have better inhibitory effect on BXM resistant strain.
- In figures, 2 & 3, viral titers are shown in different tissues in nine different treatment groups. Most of the treated groups showed only 1 mouse per group while other groups had 3-4 mice per group and showed significant reduction in viral titers in few of them. Why is the viral titer not tested for 5 mice per group that is important for the reliability of the results and for the consistency purposes?

Reviewer 3 Report
In this study authors characterised the efficacy of baloxavir acid (BXA) against highly pathogenic influenza A virus of H5N1 subtype in vivo. The study nicely complements existing research on BXA antiviral efficacy on influenza viruses by looking specifically at the H5Nx group of viruses in mammalian host. The in vivo model (mice) was chosen appropriately for the study and is relevant to research on host tropism of zoonotic H5Nx influenza viruses. The authors also assessed the efficacy of combined treatment with BXA and oseltamivir (neuraminidase inhibitor) in the same in vivo model, and this aligns in time with the recent publication looking at the same drug combination in mice infected with H1N1pdm09 strain by Park et al., 2021:Antiviral Research. There is however need for major improvement of the manuscript content, see the detailed comments below.
Major comments:
1) The English language
It’s clear from the figures what the results are but the way they are delivered in text makes it really difficult to understand. I suggest a major proofreading by English native speaker who would help in re-shaping and re-phrasing the content so that it’s more reader‑friendly. Example: Line 392-394: Results further showed that BXM monotherapy or BXM and OSP combination therapies was significantly reduced in the brain and kidney virus titers at 5, 6, and 7 dpi compared to vehicle or OSP monotherapy (Figs. 6b, 6c) COULD BE ALTERED TO Furthermore, BXM mono- or BXM/OSP combined therapies significantly reduced viral titres in brain and kidney at 5, 6, and 7 d pi as compared to vehicle or OSP treatment (Fig 6 b and c).
2) General comment for materials and methods
The language used suggests the opposite order of actions – from the end to the beginning. You should change it so that it reflects the order protocols have been carried out, e.g. first lungs were collected, then homogenized, samples processed and only THEN the cytokine and chemokine profiles were determined using ELISA. See lines 196-202 for reference.
3) Choice of virus strain for in vivo studies:
Since all tested H5Nx viruses had comparable susceptibility to BXA in vitro, the A/Hong Kong/483/1997 (H5N1) strain was selected for subsequent analyses. Why did authors decide to use this strain, given there were other contemporary strains of H5N8 or H5N6 subtypes available? Authors should justify their choice in text.
4) Combined treatment
I do not fully understand the isobologram plot and the values retrieved from it - could authors explain and expand better in text, rather than just referring plainly to previous publications?
5) Line 140 - Animal experiments
It’s not clear from the text what the total sizes of animal groups were. Provide better explanation: e.g. the starting number of animals per group was X and Y number of groups. At day 3 and 5 pi X number of animals was sacrificed for investigation of virus dissemination in the host tissues and organs. The survival rate was determined based on the remaining number of animals in each group, etc. Instead of text, you could also provide the animal experiment diagram.
6) Lines 396-398 – mutations in PA
Authors failed to mention mutations in PA following OSP treatment, as opposed to combined BXA/OSP therapy. Is it possible that the samples were mixed up or the quality of samples was poor? Or would authors suspect the unconventional mechanism for acquiring OSP‑resistance through changes in viral polymerase genes? I would like authors to expand on it in discussion. For combined treatment BXA/OSP in mice and emergence of resistant variants, please refer to the recent publication by Park et al., 2021: Antiviral Research - Baloxavir-oseltamivir combination therapy inhibits the emergence of resistant substitutions in influenza A virus PA gene in a mouse model.
Minor comments:
Lines 62-64 – Baloxavir marboxil (BXM), which is converted metabolically to its active form; baloxavir acid (BXA), is a first-in-class cap-dependent endonuclease (CEN) inhibitor that has recently been approved for clinical use in over 65 countries in 2021. – Please add reference
Line 90 – “in seasonal influenza”instead of “of seasonal influenza…”
Line 448-450 Does BXA show systemic distribution?